# Assessment of the Geo-Environmental Status of European Union Priority Habitat Type "Mediterranean Temporary Ponds" in Mt. Oiti, Greece



**Charalampos Vasilatos** [1,*] , **Marianthi Anastasatou** [1] , **John Alexopoulos** [2] ,
**Emmanuel Vassilakis** [3] , **Spyridon Dilalos** [2] , **Sofia Antonopoulou** [1] , **Stelios Petrakis** [3] ,
**Pinelopi Delipetrou** [4] , **Kyriacos Georghiou** [4] and **Michael Stamatakis** [1]

[1]   Department of Economic Geology and Geochemistry, Faculty of Geology and Geoenvironment,
      National and Kapodistrian University of Athens, Panepistiomiopolis, 15784 Zografou, Athens, Greece
[2]   Department of Geophysics-Geothermics, Faculty of Geology and Geoenvironment, National and
      Kapodistrian University of Athens, Panepistiomiopolis, 15784 Zografou, Athens, Greece
[3]   Department of Geography-Climatology, Faculty of Geology and Geoenvironment, National and
      Kapodistrian University of Athens, Panepistiomiopolis, 15784 Zografou, Athens, Greece
[4]   Department of Botany, Faculty of Biology, National and Kapodistrian University of Athens,
      Panepistiomiopolis, 15701 Zografou, Athens, Greece
*     Correspondence: vasilatos@geol.uoa.gr; Tel.: +30-210727-4664

**Abstract:** Mediterranean Temporary Ponds (MTPs) constitute priority habitat under the European Union Habitats' Directive. They are inhabited by rare species and subjected to unstable environmental conditions. Lakes and ponds act as early indicators of climate change, to which high altitude ecosystems are especially vulnerable. This study presents a full dataset of the geo-environmental parameters of such habitats (MTPs) along with their current ecological status for the first time. Furthermore, this paper aims to address the lack of basic geo-environmental background on the network of MTPs of Mt. Oiti concerning their geological, geomorphological, mineralogical and geochemical characteristics along with the pressures received from various activities. The study area is located in a mountainous Natura 2000 site of Central Greece, which hosts four MTPs. Fieldwork and sampling of water and bottom sediments were carried out during dry and wet periods between 2012 and 2014. Electrical Resistivity Tomography measurements identified synforms shaped under the ponds that topography does not always adopt them, mostly due to erosion procedures. The most significant feature, distinguishing those pond waters from any other province water bodies is the extremely low content of all studied ions (including $NO_2^-$, $NO_3^-$, $NH_4^+$, $PO_4^{3-}$, $HCO_3^-$, $SO_4^{2-}$, Al, As, B, Ba, Ca, Cd, Ce, Cl, Co, Cr, Cs, Cu, Fe, Ga, Gd, Ge, Hf, Hg, K, La, Li, Mg, Mn, Mo, Na, Ni, P, Rb, S, Sb, Se, Si, Sn, Sr, Ti, U, V, W, Zn, and Zr). MTPs water bodies are of bicarbonate dominant type, and a fresh meteoric water origin is suggested. The main pressures identified were grazing and trampling by vehicles. MTPs of Mt. Oiti were classified according to their ecological status form excellent to medium. Our results can contribute to a better understanding of the mountainous temporary ponds development in the Mediterranean environment.

**Keywords:** Mediterranean Temporary Ponds; freshwater habitats; ecology; environmental geochemistry; electrical resistivity tomography

## 1. Introduction

Temporary ponds (TPs) are globally known under different names, such as temporary aquatic habitats, seasonal wetlands, ephemeral wetlands, vernal pools, etc. [1,2], and their importance is related

to the existence of several rare and endemic species [3,4]. Temporary ponds are naturally widespread in all geographical regions and rare habitats, in contrast to their ephemeral character, are amongst the most long-term of aquatic habitats [5]. Within the EU, this habitat is mainly distributed in dry and sub-arid areas of its southern region.

European Union countries have performed an assessment of all freshwater habitats larger than 50 ha to meet the requirements of the Water Framework Directive 2000 [6]. A set of indicators and multimetric indices has been developed for the purposes of biological, hydromorhpological and physico-chemical monitoring, such as morphology, hydrology, nutrient status, thermal conditions, salinity, pollutants and priority substances. Thus, such indicators have been developed for larger water bodies and they are still lacking for smaller wetlands, especially for temporary ponds which are typically smaller than 10 ha and can be as small as 1 m$^2$. This is in contrast with the conservation value, valuable ecosystem functions and the unique biodiversity of these systems [7].

Temporary ponds in the Mediterranean regions of Europe have been studied more intensively and are under effective protection status, as a result of their identification as priority habitat (Annex I code 3170*) in the EU Directive 92/43/EEC (Habitats' Directive) [8]. Mediterranean Temporary Ponds (MTPs) are shallow water bodies with annual inundated and dry phases of varying duration and timing [2,9,10]. The main ecological characteristic of the habitat is that the autumn-winter wet (aquatic) ecophase is followed by a spring-summer dry (terrestrial) ecophase. The typical species found in them are particular in the sense that they are often dwarf, "amphibious" species which have adapted to this alternation of ecophases [1,11,12]. Thus, these seasonal water bodies, regardless of their small size, operate as biodiversity hotspots [13,14] maintaining gamma diversity and they host considerable diversity of flora and fauna species that are often rare and endemic occurring uniquely in this habitat [2,8,10–12].

The distribution of MTPs reflects the Mediterranean climate zones [1], though there is insufficient knowledge about their selective spatial distribution [12]. These temporary ponds have often been unidentified by conservationists, in contrast to permanent water ecosystems, even within protected areas [5]. More recently, it is a common concern of the scientific community that the delicately balanced hydrological regime of these temporary water bodies, might be exceptionally susceptible to various factors, including soil drainage for agriculture or urban development, high vulnerability to pollutants, lack of public awareness and climate change [5]. These ponds have been reported in various Mediterranean countries, such as Greece where they are known under the names "arolithoi", "rousies" and "kolympes" [15].

Due to the seasonality and vulnerability of the MTPs it is important to identify the biotic and abiotic parameters during the dry and the inundated ecophase, in order to provide an integrated description of these systems and understand their existence and development. Altitude, the surface area of the wet system, pond size and disturbances, such as grazing by cattle, have been found to be very important for biota [2,15,16]. Furthermore, different geological and lithological conditions [12], timing and duration of inundated and dry period [17,18], soil and water physicochemical factors [2,12,14], size and depth of the pond, water supply [2,15] and trophic status [2] result in different ecological conditions forming various biotic communities [18]. Specifically, geochemical indicators, such as pH, sediment mineralogy, concentrations of ions, nutrients and potential pollutants in the soil and water column determine the quality and allow to quantify the environmental stress impacting the ecosystem (ecological status) [7]. In terms of ecosystem functions, these ponds are important biodiversity hotspots, in the sense of species composition and biological traits [19]. Well-connected networks of ponds are vital in the provision of new climate space, and spatial planning should be encouraged in order to enable pond biota to adapt to climate change [19]. However, threats to these habitats strongly depend on specific site conditions, including pasture, expansion of cropland, forestry, hunting and touristic activities [20].

In Greece, there are 33 documented MTP sites—73% of which are located on the Aegean islands [2,15], and the rest in the mainland. According to Zacharias et al. [2], 39% of the MTPs in Greece

are located on silty and sandy formations, 17% on the calcareous substrate and the remaining 44% lays on siliceous and volcanoclastic sediments (sedimentary deposits mostly composed of rock fragments of volcanic origin). The most common threat to Greek temporary ponds relates to intensive agricultural activities, which either expand over the MTPs or pollute their water bodies with fertilizers [2]. Thorough geological studies are of high importance, since they could set up the geo-environmental baselines and their results could be used as reference points at the future (e.g., Reference [21]) in the view of climate change or prior to any conservation management actions. Moreover, the geochemical conditions that dominate in MTPs, form an individual environment, for each one, of high geochemical value, which has been little studied up to now.

This paper focuses on four small and independent MTPs of high altitude thatlie within the National Forest Park area at Mt. Oiti, Central Greece. They are the only MTPs known on Mt. Oiti and the surrounding mountain ranges of Central Greece, with the exception of three ponds on Mt. Kallidromo which have different vegetation types and host few representative flora species. The flora composition of these ponds includes a total of 83 plant species; nine of them are dominant and special to temporary ponds. These ponds include the endangered local endemic species *Veronica oetaea*, which grows at the beginning of the dry period [22]. The fauna composition includes nine specialized (one possibly endemic) crustaceans [23]. The ecological status, classified in accordance with Annex V of the Water Framework Directive [24–26], defines the quality of the structure and functioning of the pond. The current study adopts the evaluation method of Dimitriou et al. [27].

The objective of this research is the determination of the geo-environmental characteristics of the MTPs on Mt. Oiti, Central Greece. This is the first integrated study which contributes to the understanding of the ecosystems of Greek MTP areas, by investigating a wide range of parameters. Our findings can determine not only the ponds' geological background and ecological status, but can also contribute to a better understanding of the presence and fate of the temporary ponds in the Mediterranean region in general. Furthermore, the future monitoring of these geochemical factors will allow to detect changes and to help in early warning regarding qualitative and quantitative risks.

## 2. Materials and Methods

### 2.1. Study Area

The study was carried out in the four Mediterranean mountain temporary ponds of the National Forest Park of Mt. Oiti (GR2440004; Figure 1). These MTPs are of high ecological value as they host the endemic plant species *Veronica oetaea*. The National Forest Park of Mt. Oiti is a protected area of almost 7000 hectares and encompasses a network of temporary ponds of natural origin. Since 1966, the core of Mt. Oiti was declared a national park under the Royal Decree 218/7.3.1966 (Government Gazette 56/12.3.1966) and the wider area is designated a protected area under Natura 2000 Network: 'National Park of Mt. Oiti—Asopos Valley' GR2440007 based on the Directive 2009/147/EC (former Directive 79/409/EEC) and 'National Park of Mt. Oiti' GR2440004 based on the Directive 92/43/EEC.

Mt. Oiti is the fifth highest mountain of Greece with Pyrgos (2152 m) and Greveno (2114 m) being Oiti's highest peaks [28]. The small area with MTPs extends along the National Park of Mt. Oiti. The current paper includes the study of the four small and independent temporary ponds of high altitude in Louka, Livadies, Greveno and Alikaina (Figure 1; Table 1).

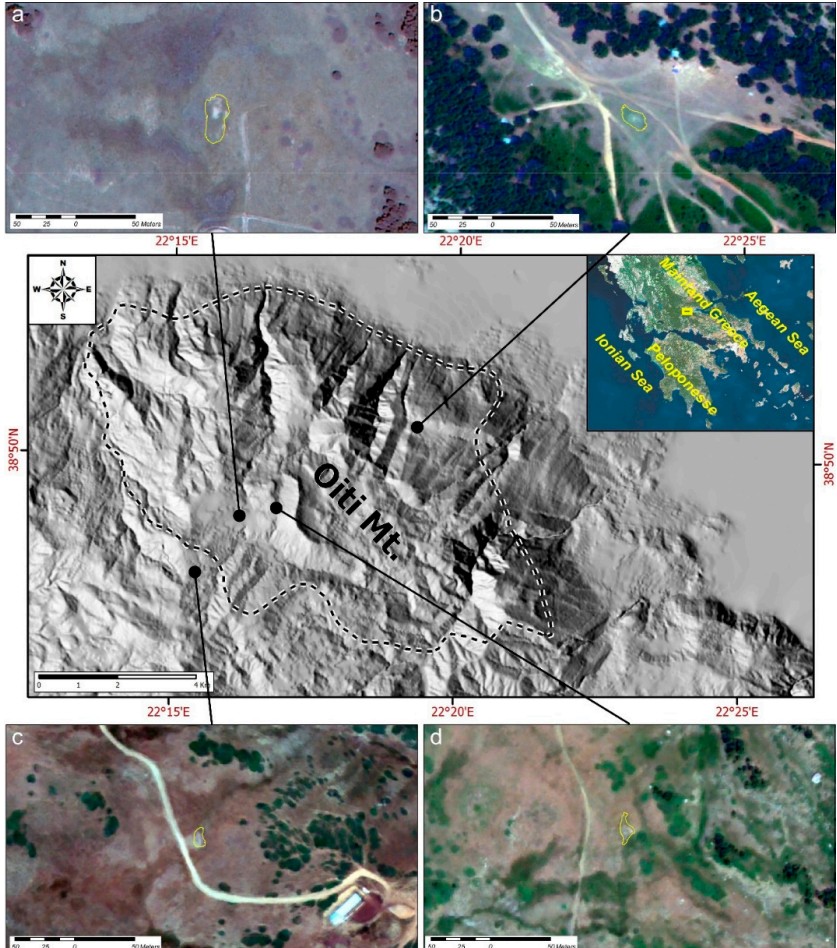

**Figure 1.** Shaded relief map (GIS environment) of Mt. Oiti and its location inset map of southern Greece. The dashed line shows the limits of the NATURA GR2440004 preservation area. The four Mediterranean Temporary Ponds (MTPs) of (**a**) Livadies, (**b**) Louka, (**c**) Alikaina, and (**d**) Greveno are also displayed in this figure on high-resolution satellite images.

**Table 1.** Mediterranean Temporary Ponds sites of Mt. Oiti, Central Greece.

| Pond Name | Abbreviation | Altitude (m a.s.l.) | Area (m$^2$) |
|-----------|--------------|---------------------|--------------|
| Livadies  | LIV          | 1815                | 607.9        |
| Louka     | LOU          | 1114                | 252.1        |
| Greveno   | GRE          | 1895                | 158.4        |
| Alikaina  | ALI          | 1919                | 112.5        |

(a.s.l. = above sea level).

The Mediterranean mountain temporary ponds of Mt. Oiti exhibit a rather regular alternation of the wet and dry ecophase with the dry ecophase starting earlier. The plant communities are mainly the ones of the Isoëtalia order, flowering in late spring or early summer. Their characteristic (diagnostic) species are mainly the annuals *Lythrum thymifolia*, *Limosella aquatica*, *Ranunculus lateriflorus*, *Myosurus minimus* and *Veronica oetaea*. The pond of Louka hosts the late summer flowering perennial species, *Mentha pulegium*. Grassland species are restricted to the margins of the ponds, with the exception of the Louka pond where grassland species intrude in the central parts of the pond. The ponds host no aquatic vegetation and the typical communities are succeeded by pioneer nitrophilous vegetation with *Polygonum arenastrum* in late summer and autumn [22].

*2.2. Geological and Geophysical Survey*

Geological fieldwork included detailed geomorphological and geological investigation and mapping, in order to fully identify the geographical and geological settings and their relation to the studied ponds. Geophysical methods were implemented in order to delineate the subsurface geological structure of the ponds, to provide information on their hydrogeological characteristics and to propose the reason for their establishment in these specific locations. The Electrical Resistivity Tomography (ERT) technique was selected as the most appropriate method [29,30] in order to investigate shallow depths in high detail. This kind of resistivity data provides a detailed image of the 2D subsurface resistivity distribution. Practically, forty-one (41) electrodes were laid out on a straight line with a constant spacing, controlled by software which automatically selects the four active electrodes used for each measurement. Therefore, based on that software, all the possible sets of four electrodes with the same spacing are selected in order to collect multiple resistivity data for both vertical and lateral resistivity investigation. The acquired resistivity data were processed with the Res2DInv Software (Geotomo) in order to generate the 2D subsurface model.

A total of ten (10) ERT profiles were carried out over these four MTPs (Figure 2), covering a total length of subsurface investigation equal to 1520 m and depth of investigation up to 10–20 m. Actually, at every pond, two or three ERT sections of different direction were carried out, during the dry seasons of 2013–2015. Collecting the ERT profiles of different directions over the ponds provide a 3D perspective concerning the resistivity distribution and, consequently, the evaluation of the subsurface geological structures.

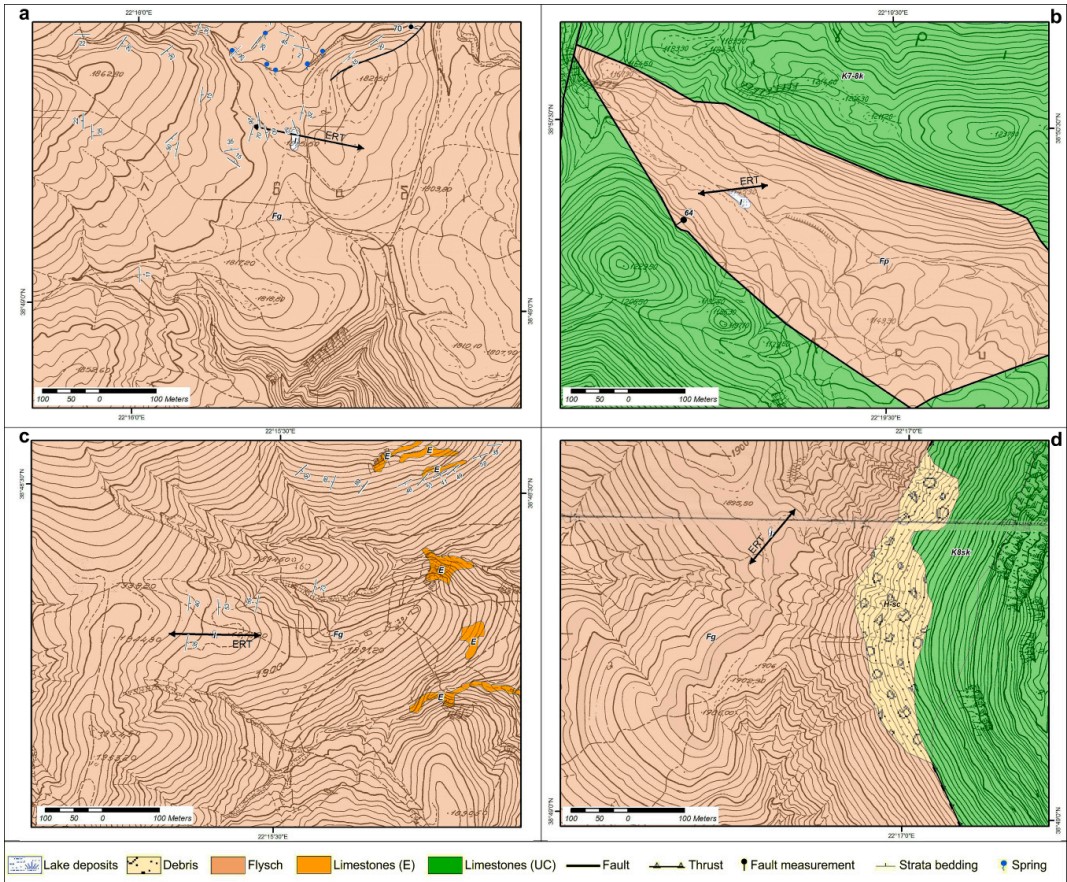

**Figure 2.** Geological maps along with the overall configuration of ERT profiles of Mt. Oiti's MTPs: (**a**) The Livadies pond, (**b**) Louka pond, (**c**) Alikaina pond, and (**d**) Greveno pond.

### 2.2.1. Pond of Livadies

The pond of Livadies (Figures 1a, 2a and 3a,b) is located in the southwest part of Mt. Oiti and lies on the flysch formation of Eastern Greece unit, which comprises of coarse sandstones, shales and sandy marls with often permeable carbonate and conglomerate intercalations [31]. The formations, on top of which the pond is deployed, are generally impermeable—especially when there is no tectonic disturbance caused by faulting or thrusting.

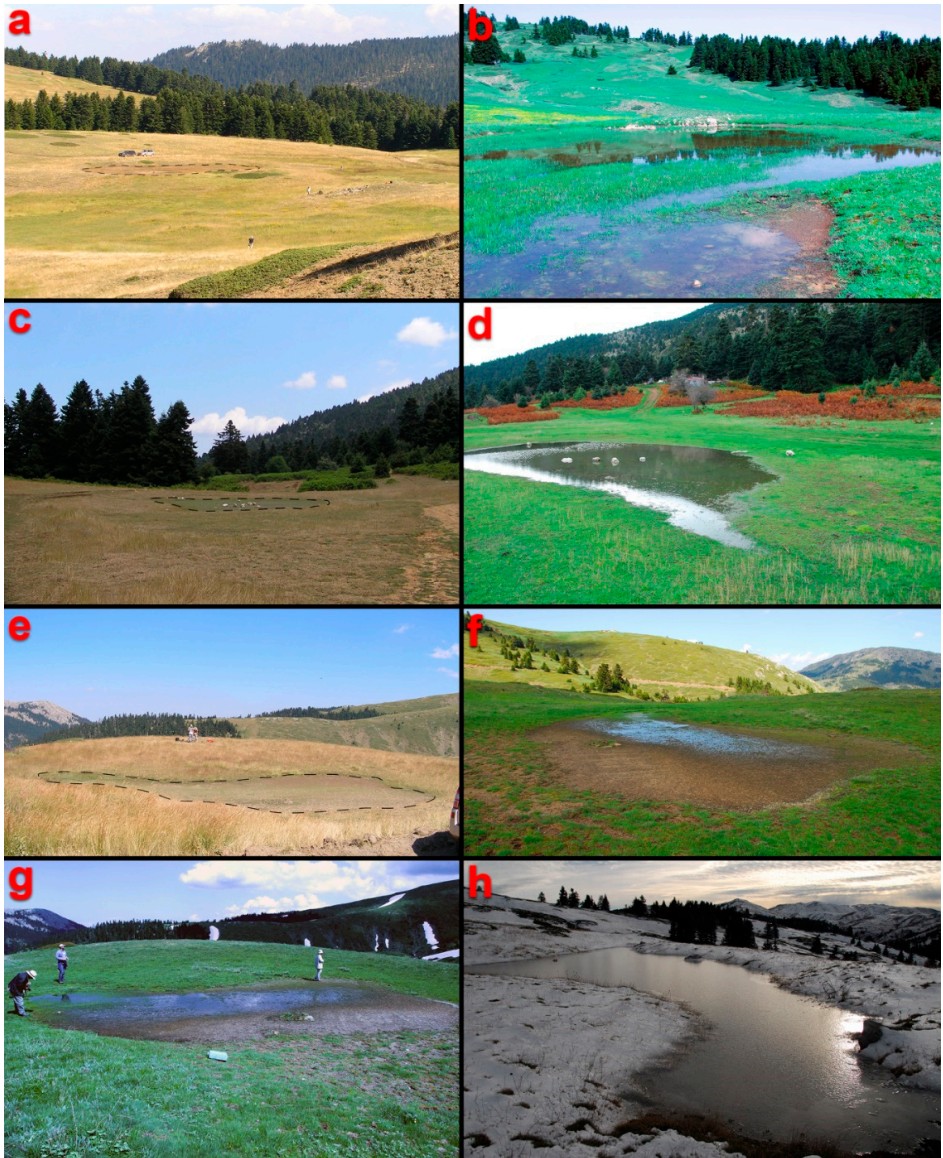

**Figure 3.** Mediterranean Temporary Ponds at Mt Oiti during dry * (left hand side) and wet (right hand side) periods: Livadies (**a**,**b**), Louka (**c**,**d**), Alikaina (**e**,**f**), and Greveno (**g**,**h**). (* dashed lines).

### 2.2.2. Pond of Louka

The Louka pond is located in the northeastern part of Mt. Oiti. The surrounding area is dominated by alpine formations which form an NW-SE trending graben with Upper Cretaceous limestones having uplifted, and Eocene flysch [32] subsided through several faults (Figures 1b, 2b and 3c,d). The water is collected on the impermeable layers of the flysch, which consists of Paleogene sediments (turbidites), shales and sandstones [33].

The endemic species of *Veronica Oetaea* was not recorded in the area. The European Environmental Agency, under EU Article 17 of the Habitats Directive in 2007, states that the possible exploitation of slopes rich in bauxite, may contribute to the absence of habitats.

### 2.2.3. Pond of Alikaina

The temporary pond Alikaina is formed on a morphological ridge almost identical to that of Livadies (Figures 1c, 2c and 3e,f). The basement consists of the Eastern Greece unit flysch formation [31], which is folded on an NNE-SSW trending axis.

### 2.2.4. Pond of Greveno

The small temporary pond of Greveno is formed at about 1 km to the east of Livadies, but at a slightly higher elevation (Table 1) on top of the same flysch formation with intercalations of permeable and impermeable layers, as mentioned above. More to the east, a thick layer of limestones is found that was thrusted up, on top of the flysch, resulting in a 200 m high morphological discontinuity (Figures 1d, 2d and 3g,h). The tectonic contact is covered by debris consisting of large carbonate boulders, which are pretty often being dispatched and roll down to lower elevations.

### 2.3. Physical Characteristics

From 2012 to 2014, a multidisciplinary study was conducted on MTPs based on fieldwork, remote sensing (by processing historical aerial photographs and contemporary high spatial/spectral resolution satellite images) and topographic surveying. Several characteristics for each pond were measured, including perimeter and relief of the surrounding areas, as well as the pond bottoms with the use of high accuracy Real-Time-Kinematics-GPS equipment. A morphometric parameter, Shape Index (SI) [34], was used to define pond shape complexity:

$$SI_i = \frac{P_i}{2\sqrt{\pi \alpha_i}}$$

where P is the pond perimeter (m), and $\alpha$ is the pond area (m$^2$). SI is 1 when the pond is circular and increases as the pond becomes more complex. Moreover, SI has no units, but is related linearly to the richness of plant species at a pond or lake. Furthermore, according to Bagella et al. [35], the higher perimeter to area ratio enhances the biogeochemical functions.

### 2.4. Sampling and Analytical Procedures

During the years 2012, 2013 and 2014 bottom sediments and water were collected from the temporary ponds of the National Forest Park of Mt. Oiti (Table 1). Sediments and water sampling were carried out over two different time periods, autumn and spring, in order to ensure the representativeness of the data, compare them for the same general conditions (same periods) and reflect disturbances by seasonal factors (different periods). In each pond, the sediments sampling was carried out at the bottom of the outer part of the pond. Sampling was performed from designated sampling points with a maximum uppermost of 15 cm and side inspection of plant communities.

### 2.4.1. Bottom Sediments

Bottom sediments of the ponds were collected in 2 L plastic bags. Before the experimental procedure, the coarse organic matter was removed from the samples. Bottom sediment samples were air-dried, deflocculated and homogenized, removing the floristic presence. The granulometric characterization was conducted according to Folk's methodology and classification [36], and all the statistical parameters were calculated by the software GRADISTAT v.8 [37]. The mineralogical analysis was carried out by using an X-Ray Diffractometer of Siemens D 5005 type, equipped with a copper tube and a graphite monochromatographe, in combination with the DIFFRACplus software package

at the laboratories of NKUA, Faculty of Geology and Geoenvironment. The $SiO_2$, $Al_2O_3$, $Fe_2O_3$, MgO, CaO, $Na_2O$, $K_2O$, $TiO_2$, $P_2O_5$, MnO and $Cr_2O_3$ concentrations were analyzed using the ICP AES method and the trace elements (Ag, As, Au, Ba, Be, Bi, Cd, Ce, Co, Cs, Dy, Er, Eu, Ga, Gd, Hf, Hg, Ho, La, Lu, Mo, Nb, Nd, Ni, Pb, Pr, Rb, Sb, Sc, Se, Sm, Sn, Sr, Ta, Tb, Th, Tl, Tm, U, V, W, Y, Yb, Zn, Zr) were analyzed by the ICP MS method at Bureau Veritas (ex ACMELabs) analytical laboratories in Canada (Supplementary Table S1). The organic matter (OM) was measured through dry combustion (humidity 105 °C/24 h, burning 380 °C/6 h) using a muffle furnace at the laboratories of National and Kapodistrian University of Athens (NKUA), Faculty of Geology and Geoenvironment. The geochemical interpretation of the trace elements was carried out by using plug-in modules of GCDKit 4.0 software.

### 2.4.2. Water

The measurements of the pH, redox potential (Eh), conductivity and Total Dissolved Solids (TDS) values were performed in situ using a portable Multiparameter Analyzer (Consort 561). Water samples were filtered through 0.4 μm membrane filters, collected and stored in polyethylaine containers. They were divided into two groups; the first group was perseved for anions' measurements, the second was acidified by addition of concentrated $HNO_3$ for cations' measurements, and then they were all stored at cooling conditions (~4 °C) into a portable refrigerator.

Chemical analyses for $NO_2^-$, $NO_3^-$, $NH_4^+$, $PO_4^{3-}$, $HCO_3^-$, $SO_4^{2-}$ were performed using a HACH DR/4000 spectrometer. The estimation detection limits of the methods were 0.003 mg·$L^{-1}$, 0.05 mg·$L^{-1}$, 0.09 mg·$L^{-1}$, 0.05 mg·$L^{-1}$, 5 mg·$L^{-1}$ and 3 mg·$L^{-1}$, respectively. The trace elements (Ag, Al, As, Au, B, Ba, Be, Bi, Br, Ca, Cd, Ce, Cl, Co, Cr, Cs, Cu, Dy, Er, Eu, Fe, Ga, Gd, Ge, Hf, Hg, Ho, In, K, La, Li, Lu, Mg, Mn, Mo, Na, Nb, Nd, Ni, P, Pb, Pd, Pr, Pt, Rb, Re, Rh, Ru, S, Sb, Sc, Se, Si, Sm, Sn, Sr, Ta, Tb, Te, Th, Ti, Tl, Tm, U, V, W, Y, Yb, Zn, Zr) were analyzed in the acidified portion of the samples with the ICP MS method at Bureau Veritas (ex ACMELabs) analytical laboratories in Canada (Supplementary Table S2).

GW CHART software, version 1.29.0.0 was used to construct Piper diagrams and identify the hydrochemical facies.

## 3. Results

### 3.1. Geological and Geophysical Survey

The surface and subsurface geological setting of the extended area of the MTPs network, located in Mt. Oiti, was considered for the multifactorial correlation of the abiotic parameters results.

During the field mapping at the surrounding area of the Livadies pond, many tectonic structures were identified especially at the north of the pond (Figures 1a, 2a and 3a,b). The flysch formation seems to be folded as the dipping direction of the strata varies and changes every few meters. The orientation of the folding axes trends is mainly NNE-SSW, and the bed dipping reaches up of 70°. High bedding dips are observed only west of the pond. The pond is settled on a structural synform created by the flysch impermeable layers. A number of low yield springs were observed along a valley north of the pond area, which discharges the water from the permeable layers of the flysch formation. Any groundwater circulation is obstructed by an NE-SW trending fault. The footwall of this fault hosts the pond whilst the groundwater appears at the hanging wall. Taking into account the above, it is more than certain that the water of the pond is a result solely of precipitation.

At the area of the Livadies pond (Figure 4)—especially at the first 10 m in depth—a structure of two geoelectrical layers has been identified. A geoelectrical layer of relatively lower resistivity values, below 80 Ohm.m (bluish colors), has been detected forming small and smooth 'geoelectrical' folds in the subsurface. This measurement is in agreement with the observations made during the geological field-mapping of the present study and the existence of the flysch formation (alterations of different lithologies).

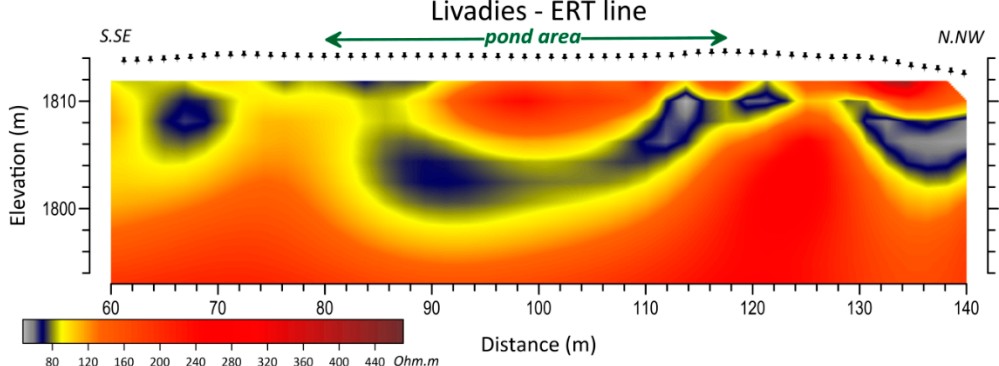

**Figure 4.** Part of the electrical resistivity tomography (ERT) line across the Livadies pond.

At the area of the Louka pond (Figure 5), the Electrical Resistivity Tomography outlines a quite conductive geoelectrical layer (<100 Ohm·m) at the first meters, overlying a more resistant formation. This may correspond to a small local basin filled with a few meters of post-alpine sediments of lower resistivity.

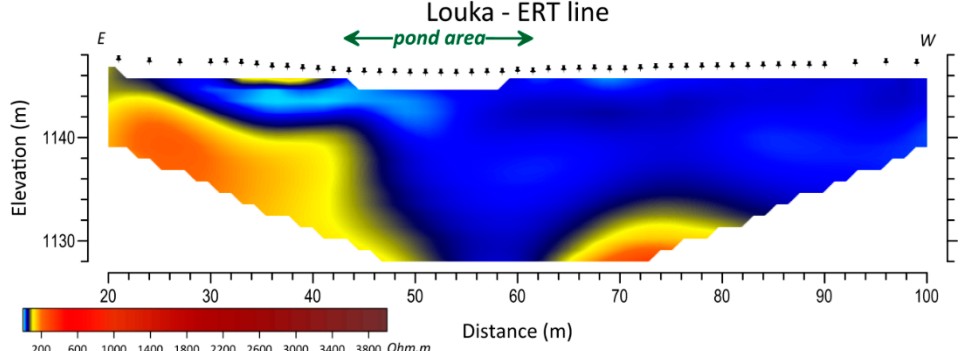

**Figure 5.** Part of the ERT line on the Louka pond.

The overall structure of the Alikaina pond is almost identical to the one that is described for the Livadies pond (see Section 2.2.1), even though the altitude is significantly higher. The Eocene limestone intercalations, which are observed at the surrounding area, are part of the same folding structures as their bedding is in agreement to the flysch [31]. Moreover, the water body is collected above a synform. Regarding the ERT measurements at the area of the Alikaina pond (Figure 6), they have been determined similar to the resistivity values of Louka. The resistivity values increase progressively from small depths to greater ones. In the extended area of the pond, a conductive geoelectrical layer (<100 Ohm·m) has been identified to a maximum depth of almost 8 m. Underlying, a more resistant formation has been delineated.

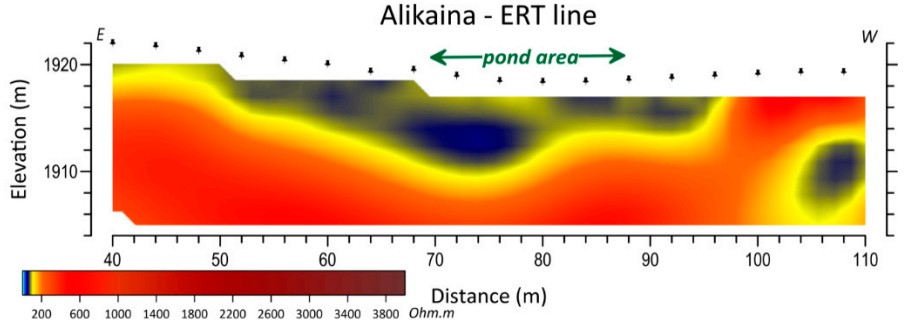

**Figure 6.** Part of the ERT line on the Alikaina pond.

Regarding resistivity values, the geoelectrical image of the Greveno pond area (Figure 7) is similar to the one of Livadies (Figure 4). In both cases, alternations of two geoelectrical layers have been determined. A quite conductive layer (<100 Ohm·m) of almost two meters thickness has been revealed below the pond. A more resistant layer of 4–5 m has been detected below that, and a more conductive one was identified at even greater depths.

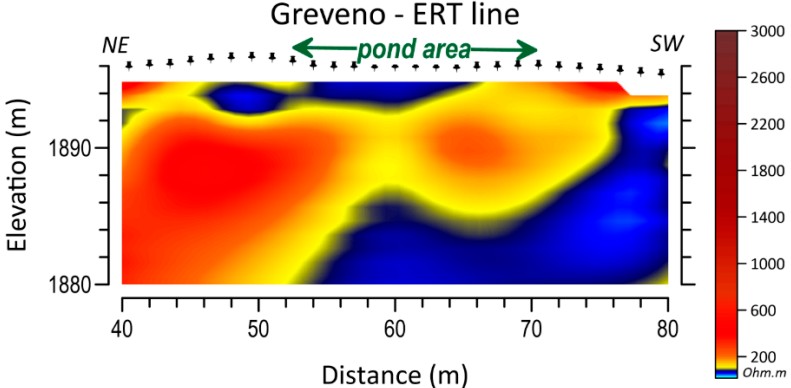

**Figure 7.** Part of the ERT line across the Greveno pond.

### 3.2. Physical Characteristics

The network of MTPs is located at Mt. Oiti, the respective ponds areas range from 112 m$^2$ to 608 m$^2$ and the perimeters from 46 m to 112 m. The four pond sites are of various sizes and shapes (Figures 1 and 8), including circular to linear (Alikaina pond; Figure 1c; SI = 1.52), near-linear (Livadies and Louka ponds; Figure 1a and SI = 1.63; Figure 1b and SI = 1.65, respectively) and convoluted (Greveno pond; Figure 1d and SI = 2.31). Pool areas and perimeters were not significantly different between the Greveno and Alikaina ponds, but were at a slightly higher altitude at the Louka pond and significantly higher at the Livadies pond. Figure 3 illustrates examples of typical pools within each site. However, shape complexity was notable higher at Greveno, compared to the other three sites, and it should be noted that Alikaina displayed the lowest SI. Mean pond depths range between 5 and 21 cm, with the Livadies pond being significantly deeper than the other three ponds (Table 2). The altitude of the study area varies between 1114 m (Louka pond) and 1919 m (Alikaina pond). Notably, the Louka pond, is situated at the lowest altitude compared to the other three ponds. The main pressures identified during fieldwork observations for those MTPs were grazing and trampling by vehicles.

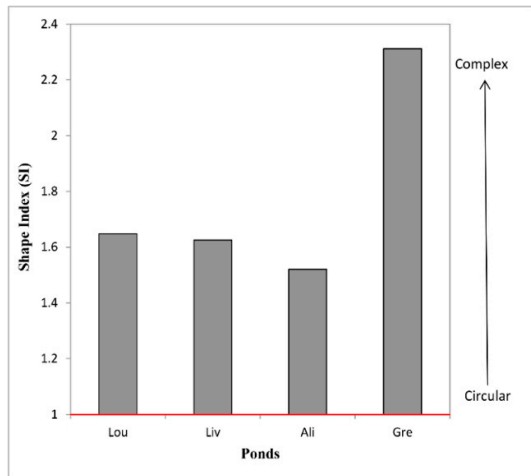

**Figure 8.** Shape Index (SI) for each MTP in Mt. Oiti area, indicates that shape complexity (see Figure 1) conforms to pond's foremost biogeochemical functions.

**Table 2.** Measured and calculated physical parameters of MTPs.

|  | **Livadies** | **Louka** | **Greveno** | **Alikaina** |
|---|---|---|---|---|
| Area (m$^2$) | 607.9 | 252.1 | 158.4 | 112.5 |
| Perimeter (m) | 111.4 | 72.3 | 67.8 | 46.4 |
| Shape Index (SI) | 1.63 | 1.65 | 2.31 | 1.52 |
| Mean pond depth (cm) | 20.4 | 8 | 5.3 | 5 |
| Bottom sediments granulometry | Gravelly Mud | Sandy Mud | Sandy Mud | Mud |

### 3.3. Bottom Sediments

According to the granulometrical analysis of the samples (Figure 9) by Folk's categorization, the bottom sediments from the Alikaina pond are characterized as mud, from the Louka pond as gravelly mud and from the Livadies and Greveno pond as slightly gravelly, sandy mud and gravelly mud, respectively.

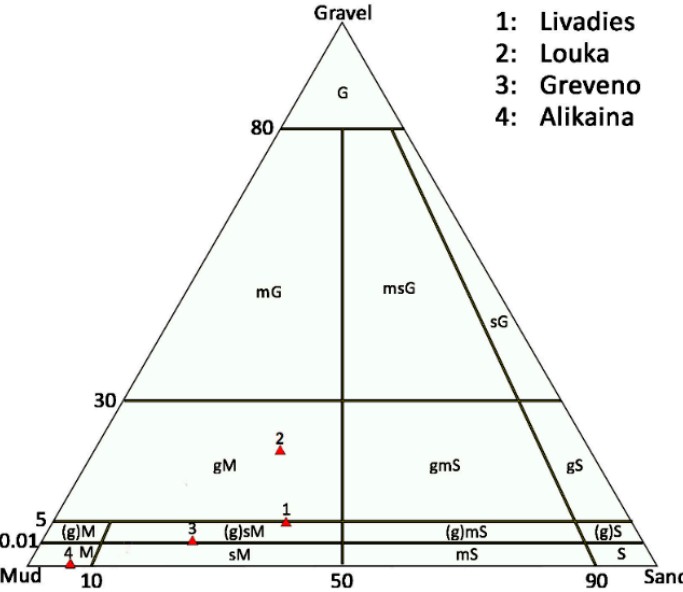

**Figure 9.** Schematic representation of bottom sediments samples granulometry from Oiti's temporary ponds during December 2014, according to the Folk classification scheme.

The results of the qualitative mineralogical analysis are shown in Table 3. The predominant mineral phase in all bottom sediment samples is quartz. Moreover, albite, chlorite, iron-rich smectites and clay minerals (illite) occur frequently. Small amounts of K-feldspars (mainly orthoclase), sepiolite and hornblende were detected in some samples.

**Table 3.** Qualitative and semi-quantitative determination of the mineralogical phases in the bottom sediments samples by powder X-Ray Diffraction spectroscopy. For the pond abbreviations, see Table 1.

|  | **LIV** | **LOU** | **GRE** | **ALI** |
|---|---|---|---|---|
| **Quartz** | MJ | MJ | MJ | MJ |
| **Alkali Feldpars** | MD | TR/MD | MD | MD |
| **Illite** | TR | TR | TR/MD | MD |
| **Smectite** | TR/MD | TR/MD | TR | tr |
| **Chlorite** | TR | TR | TR | TR |
| **Sepiolite** | tr | tr | tr |  |
| **Hornblende** | tr |  |  |  |

(MJ = Major phase, MD = medium phase, TR/tr = minor/trace phase, respectively and Alkali Feldpars = mainly albite and rarely orthoclase).

The chemical analysis of the bottom sediment samples is presented in Table 4. Despite the fact that all samples exhibit slightly increased iron content, there is no significant presence of any iron oxide mineral phase. These high iron contents may be related to iron-rich clay minerals, such as iron-rich smectite and chlorite. The Louka pond exhibits the highest iron value, whereas the predominance of clay minerals is related to low $SiO_2$ content. Potassium content is higher than sodium and may be related to the presence of illite, as K-feldspar (orthoclase) occurs rarely. CaO content is low (0.6%), as carbonate minerals and plagioclase are absent. MgO content is attributed to the presence of chlorite, secondarily to other magnesium bearing minerals and with minor relevance to amphiboles (e.g., edenite). $P_2O_5$ content is twofold in the Greveno and Alikaina ponds in relation to Louka and Livadies. OM content ranged from 3.24 at the Livadies ponds to 9% at the Greveno pond.

**Table 4.** Representative major and trace elements analytical results of bottom sediments samples from Oiti MTPs (OM = Organic matter). For the pond abbreviations, see Table 1.

| | | | | | | Ponds | | | | | | | | |
|---|---|---|---|---|---|---|---|---|---|---|---|---|---|---|
| **Variable** | **LIV** | **LOU** | **GRE** | **ALI** | | **LOU** | **LIV** | **ALI** | **GRE** | | **LOU** | **LIV** | **ALI** | **GRE** |
| % | | | | | $\mu g \cdot g^{-1}$ | | | | | $\mu g \cdot g^{-1}$ | | | | |
| $SiO_2$ | 62.43 | 52.4 | 54.96 | 57.93 | As | 8 | 15 | 8 | 4 | Ni | 144 | 77 | 129 | 80 |
| $Al_2O_3$ | 14.65 | 17.98 | 14.92 | 15.21 | Ba | 383 | 318 | 315 | 333 | Pb | 69 | 20 | 24 | 24 |
| $Fe_2O_3$ | 7.1 | 7.5 | 6.11 | 7.39 | Bi | 0 | 0 | 0 | 0 | Pr | 8 | 6 | 6 | 6 |
| MgO | 2.38 | 2.35 | 2.3 | 2.83 | Ce | 80 | 61 | 62 | 62 | Rb | 140 | 100 | 120 | 113 |
| CaO | 0.42 | 0.54 | 0.57 | 0.42 | Co | 27 | 26 | 28 | 22 | Sc | 20 | 15 | 16 | 15 |
| $Na_2O$ | 1.5 | 0.59 | 1.37 | 1.42 | Cs | 9 | 4 | 6 | 5 | Sm | 5 | 4 | 5 | 5 |
| $K_2O$ | 2.18 | 2.45 | 2.36 | 2.8 | Cu | 39 | 35 | 42 | 32 | Sr | 58 | 72 | 63 | 74 |
| $TiO_2$ | 0.86 | 0.98 | 0.8 | 0.81 | Dy | 5 | 4 | 4 | 4 | Ta | 1 | 1 | 1 | 1 |
| $P_2O_5$ | 0.15 | 0.19 | 0.3 | 0.38 | Er | 3 | 2 | 2 | 2 | Tb | 1 | 1 | 1 | 1 |
| MnO | 0.05 | 0.06 | 0.06 | 0.08 | Eu | 1 | 1 | 1 | 1 | Th | 12 | 9 | 10 | 10 |
| $Cr_2O_3$ | 0.024 | 0.036 | 0.021 | 0.036 | Ga | 22 | 18 | 18 | 18 | U | 3 | 3 | 2 | 3 |
| C | 1.22 | 2.4 | 4.67 | 2.2 | Gd | 5 | 4 | 4 | 4 | V | 164 | 131 | 141 | 125 |
| S | <0.02 | <0.02 | <0.02 | <0.02 | Hf | 5 | 6 | 5 | 5 | Y | 24 | 22 | 21 | 21 |
| OM | 3.24 | 5.99 | 9 | 4.78 | La | 39 | 27 | 29 | 29 | Yb | 3 | 3 | 2 | 2 |
| | | | | | Nb | 15 | 13 | 12 | 12 | Zn | 99 | 92 | 114 | 105 |
| | | | | | Nd | 30 | 23 | 23 | 24 | Zr | 162 | 229 | 180 | 167 |

Cr, Co, Ni and V present slightly high values and most likely have the same origin as iron. Pb concentration in the pond at Louka is relatively higher than the other ponds. Sr values are quite low, due to the absence of calcareous minerals and plagioclase and Rb concentrations are low due to the trace content of K-feldspars. All the other trace elements do not present any significant concentrations.

*3.4. Water*

The physicochemical parameters, the concentration and the chemical composition of the major ions of water samples from the MTPs of Mt. Oiti are presented in Table 5.

The pH values vary between 5.8 and 7.6 and are within the normal range for precipitation and streams' water. The recorded values of TOC, $NO_3^-$ and $NO_2^-$ are quite low. However, during the dry period sampling of 2013, nitrates were slightly high.

**Table 5.** Physicochemical characteristics and chemical composition of the water samples from Oiti MTPs area (bdl: Below detection limit). For the pond abbreviations, see Table 1.

| | | DRY PERIOD | | | | | | | | | WET PERIOD | | | | |
| | | 2012 | | 2013 | | | 2014 | | | | 2013 | | | | 2014 | |
| Variable | Units | LIV | LOU | LIV | LOU | GRE | LIV | LOU | GRE | ALI | LOU | LIV | GRE | ALI | LIV | GRE |
|---|---|---|---|---|---|---|---|---|---|---|---|---|---|---|---|---|
| T | °C | 12.1 | 11.9 | 13.8 | 12.9 | 14.4 | 10.1 | 10.3 | 11.2 | 9.5 | 21.9 | 20.0 | 25.5 | 21.0 | 19.7 | 22.5 |
| pH | | 7.0 | 7.1 | 6.5 | 6.4 | 6.1 | 7.6 | 7.5 | 7.4 | 8.2 | 7.1 | 7.1 | 7.1 | 7.0 | 5.8 | 6.1 |
| Eh | mV | −11 | −16 | 17.3 | 22.8 | 39.6 | −37 | −20 | −39 | −46 | −16 | −16 | −16 | −11 | 63 | 47 |
| Conductivity | µS | 22.4 | 56.2 | 130 | 203 | 323 | 21.4 | 43 | 55.9 | 21.2 | 64.5 | 40.5 | 84.2 | 52.2 | 50.1 | 62 |
| TDS | mg·L$^{-1}$ | 11.2 | 28.7 | 69 | 108 | 176 | 11 | 23.3 | 29.8 | 11.6 | 31.6 | 22.1 | 42.1 | 26.1 | 26.2 | 33.1 |
| TOC | mg·L$^{-1}$ | 0.9 | bdl | 0.5 | 0.5 | 0.5 | 0.5 | 0.5 | 0.6 | 0.5 | bdl | bdl | bdl | bdl | 0.5 | 0.5 |
| NO$_3$$^-$ | mg·L$^{-1}$ | bdl | bdl | 17.26 | 32.76 | 36.30 | 7.08 | 5.31 | 5.31 | 5.75 | bdl | bdl | bdl | bdl | 5.75 | 2.66 |
| NO$_2$ | mg·L$^{-1}$ | bdl | bdl | bdl | bdl | bdl | bdl | bdl | bdl | bdl | bdl | bdl | bdl | bdl | bdl | bdl |
| NH$_3$ | mg·L$^{-1}$ | 2.67 | 0.12 | 2.55 | 1.34 | 0.24 | 0.10 | 0.12 | 0.10 | 0.06 | 0.12 | 1.09 | 0.12 | 0.12 | 0.64 | 0.77 |
| PO$_4$$^{3-}$ | mg·L$^{-1}$ | 0.09 | 0.53 | 0.1 | 0.2 | 0.1 | 0.1 | 0.1 | 0.1 | 0.2 | 0.35 | 0.45 | 0.25 | 0.95 | 0.60 | 0.30 |
| Cl$^-$ | mg·L$^{-1}$ | bdl | bdl | bdl | bdl | bdl | 0.5 | bdl | 0.5 | 0.5 | bdl | bdl | bdl | bdl | bdl | bdl |
| HCO$_3$$^-$ | mg·L$^{-1}$ | 135 | 46 | 45.9 | 51.7 | 49.7 | 59 | 43.3 | 59 | 48.9 | 70 | 46 | 56 | 69 | 44.7 | 43 |
| SO$_4$$^{2-}$ | mg·L$^{-1}$ | 8 | <3 | 7.5 | 9.9 | 9.9 | 12.3 | 7.5 | 12.3 | 9.9 | 10 | 8 | 8 | 8 | 7.5 | 7.5 |
| Al | µg·L$^{-1}$ | 9 | 79 | 38 | 120 | 82 | 78 | 128 | 77 | 139 | 36 | 71 | 68 | 45 | 116 | 193 |
| As | µg·L$^{-1}$ | bdl | bdl | bdl | bdl | bdl | bdl | bdl | bdl | bdl | 1 | bdl | 1 | 1 | 1 | 0.6 |
| B | µg·L$^{-1}$ | bdl | 10 | bdl | 6 | 6 | bdl | bdl | bdl | bdl | 6 | 9 | 28 | 28 | 14 | 9 |
| Ba | µg·L$^{-1}$ | 23 | 4 | 9 | 6 | 12 | 6 | 3 | 13 | 5 | 4 | 7 | 13 | 12 | 7 | 6 |
| Br | µg·L$^{-1}$ | bdl | bdl | bdl | bdl | 6 | bdl | 5 | 8 | bdl | 7 | 9 | 13 | 21 | 7 | bdl |
| Ca | mg·L$^{-1}$ | 39.4 | 3.1 | 2.9 | 5.3 | 4.3 | 2.5 | 3.0 | 8.2 | 4.1 | 2.9 | 6.9 | 12.2 | 12.6 | 2.4 | 1.7 |
| Cd | µg·L$^{-1}$ | bdl | bdl | bdl | bdl | bdl | bdl | bdl | bdl | bdl | 0.1 | 0.1 | 0.2 | 0.1 | bdl | 0.2 |
| Cl | mg·L$^{-1}$ | 2.0 | 2.0 | 1.0 | 1.0 | 1.0 | 1.0 | bdl | 2.0 | 1.0 | 1.0 | 1.0 | 1.0 | 5.0 | bdl | bdl |
| Co | µg·L$^{-1}$ | bdl | 0.03 | 0.04 | 0.22 | 0.06 | 0.06 | 0.07 | 0.04 | 0.16 | 0.09 | 0.15 | 0.41 | 0.22 | 0.07 | 0.21 |
| Cr | µg·L$^{-1}$ | bdl | bdl | 1 | 1 | 1 | 1 | 1 | 1 | 1 | 1 | 1 | 1 | 1 | 1 | 1 |
| Cu | µg·L$^{-1}$ | 1 | 3 | 3 | 2 | 4 | 3 | 2 | 3 | 2 | 5 | 7 | 7 | 7 | 4 | 4 |
| Fe | µg·L$^{-1}$ | 13 | 59 | 47 | 92 | 75 | 52 | 100 | 70 | 131 | 56 | 69 | 88 | 48 | 69 | 325 |
| K | mg·L$^{-1}$ | 0.4 | 3.2 | 0.9 | 1.9 | 1.1 | 1.7 | 0.8 | 2.0 | 1.0 | 1.7 | 1.3 | 3.9 | 6.5 | 1.4 | 0.7 |
| Mg | mg·L$^{-1}$ | 2.1 | 1.0 | 0.7 | 0.8 | 1.1 | 0.5 | 0.5 | 1.4 | 0.6 | 0.6 | 1.1 | 1.6 | 2.2 | 0.4 | 0.3 |
| Mn | µg·L$^{-1}$ | bdl | 1 | 1 | 9 | 2 | 1 | 2 | 1 | 4 | 5 | 4 | 39 | 10 | 1 | 2 |
| Mo | µg·L$^{-1}$ | bdl | bdl | bdl | bdl | bdl | bdl | bdl | bdl | bdl | bdl | bdl | bdl | bdl | bdl | bdl |

**Table 5.** *Cont.*

| Variable | Units | DRY PERIOD | | | | | | | | | WET PERIOD | | | | |
|---|---|---|---|---|---|---|---|---|---|---|---|---|---|---|---|
| | | 2012 | | 2013 | | | 2014 | | | | 2013 | | | | 2014 | |
| | | LIV | LOU | LIV | LOU | GRE | LIV | LOU | GRE | ALI | LOU | LIV | GRE | ALI | LIV | GRE |
| Na | mg·L$^{-1}$ | 2.5 | 0.4 | 0.8 | 0.6 | 1.4 | 0.8 | 0.2 | 1.6 | 0.4 | 0.6 | 1.4 | 0.9 | 2.6 | 0.8 | 1.0 |
| Ni | µg·L$^{-1}$ | bdl | 1 | 1 | 2 | 1 | 1 | 1 | 1 | 2 | 2 | 5 | 5 | 3 | 2 | 7 |
| P | µg·L$^{-1}$ | 35 | 226 | 44 | 72 | 55 | 29 | 39 | 40 | 67 | 191 | 100 | 406 | 144 | 249 | 130 |
| Pb | µg·L$^{-1}$ | bdl | 1 | bdl | bdl | 1 | bdl | 1 | 1 | bdl | 1 | 1 | 1 | bdl | bdl | 2 |
| Rb | µg·L$^{-1}$ | bdl | bdl | bdl | 1 | 1 | bdl | bdl | bdl | bdl | 1 | 1 | 2 | bdl | 1 | 1 |
| S | mg·L$^{-1}$ | 2.0 | <1 | 2.0 | 3.0 | 3.0 | 3.0 | 2.0 | 4.0 | 3.0 | 2.0 | 2.0 | 2.0 | 3.0 | 2.0 | 2.0 |
| Si | µg·L$^{-1}$ | 2977 | 390 | 887 | 356 | 998 | 866 | 290 | 1074 | 237 | 515 | 876 | 496 | 783 | 139 | 1232 |
| Sn | µg·L$^{-1}$ | bdl | bdl | 0.06 | bdl | bdl | bdl | bdl | bdl | bdl | 0.07 | 0.11 | 0.06 | bdl | bdl | Bdl |
| Sr | µg·L$^{-1}$ | 48 | 5 | 8 | 14 | 16 | 8 | 5 | 25 | 8 | 7 | 21 | 22 | 23 | 7 | 5 |
| V | µg·L$^{-1}$ | bdl | bdl | bdl | 1 | bdl | bdl | bdl | bdl | 1 | bdl | 1 | 2 | 1 | 2 | 1 |
| Y | µg·L$^{-1}$ | bdl | bdl | bdl | bdl | bdl | bdl | bdl | bdl | bdl | bdl | bdl | bdl | bdl | bdl | bdl |
| Zn | µg·L$^{-1}$ | 4 | 8 | 4 | 6 | 9 | 4 | 6 | 8 | 4 | 12 | 15 | 9 | 17 | 7 | 5 |

## 4. Discussion

Temporary ponds, despite their small size and simple community structure, are often considered as early warning systems of the impacts from the long-term variations to larger-scale aquatic systems [11]. Moreover, the factor altitude is critical for the threatened dwarf plant species of *Veronica oetaea* (Figure 10), as it is restricted to the high-altitude ponds of Mt. Oiti, but not to the lower ones, such as the Louka pond. According to Salerno et al. [38], since the 1980s there is a trend of the lower elevation ponds (2500 m a.s.l.) to disappear or to reduce their surface area.

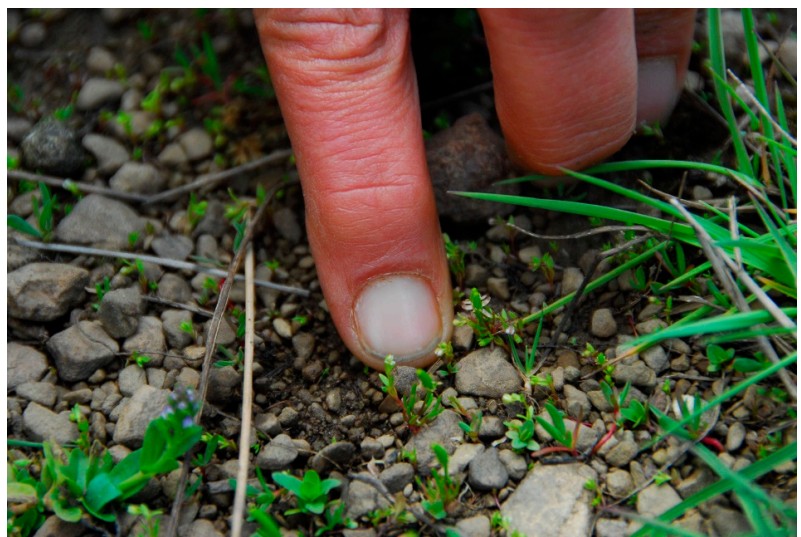

**Figure 10.** The plant species *Veronica oetaea* (May 2013). *Veronica oetaea* requires the alteration of wet and dry period and appears at the beginning of the dry period in late spring [22].

The geophysical research results in combination with the satellite images processing, led to the determination of the surface and subsurface structure of the Mt. Oiti's MTPs. Our interpretation of the ERT profiles acquired across the pond of Livadies, displayed a synform of high resistance rock that has been outlined exactly under the pond itself (Figure 4), which probably contributes to the formation of the temporary pond [29]. On the contrary, the topography does not appear to have adopted the synform, neither the erosion procedures have altered the open surface geomorphology. Overall, the water is collected on a morphological ridge. The results of the ERT sections at the pond of Louka (Figure 5) have clearly reveal displacements of high resistivity basement rocks (alpine basement), which seem to be synthetic structures to the westernmost normal fault that forms the Louka graben. This basement structure is covered with sediments, characterized by lower resistivity. This pond differs from the one described at Livadies, as the topography is clearly related to a subsided basin like water collection landform and not a morphological ridge. Regarding the pond of Alikaina (Figure 6), a geoelectrical synform has also been identified from the ERT profiles, but with a more progressive distribution of resistivity values, increasing from small depths to greater ones. No connection to any groundwater was identified. At the Greveno pond, the ERT survey (Figure 7) has also revealed a high resistivity synform outlined partly under the pond. As a result, the ponds seem to be sealed from any groundwater aquifer. Furthermore, with the exception of the Louka pond, the topography prevents any runoff of surface waters to the ponds. Therefore, the origin of the water of those ponds is confined to precipitation and, thus, water chemistry may be controlled by biological activities and water–rock interaction in the ponds' bottom sediments.

Abiotic indicators, such as physical and chemical indicators, are specific of soil and water conditions and can be used to specify various disturbances impacting the wetland ecosystem [7]. In a permanent pond or lake, the OM accumulates at the bottom of the habitat, while in a temporary pond, there is no sediment build-up every year at its bottom [5]. In temporary ponds, most of the OM

originates from the wet phase and oxidizes during the dry phase, as OM turnover is greatly correlated with redox conditions in wetlands sediments [39]. Due to their small size and water volume, ponds are particularly sensitive to climate conditions as both the organic and inorganic compounds of their sediments are subject to oxidative aerial processes during the dry periods resulting in quite low OM values. This aerial oxidation of the OM contributes to the timelessness of these ponds preventing filling by organic residues. Concerning the MTPs of Mt. Oiti, the correlation between SI and OM (Supplementary Table S3), which is directly related to the flora presence, exhibits a strong correlation coefficient of 0.89. TDS concentration is extremely poor, regardless of seasonality.

The bottom sediments granulometry of the ponds varies around the MTPs area. Between SI and gravel content, there is a positive correlation coefficient of 0.79. The mineralogy of the bottom sediments of the ponds does not display any significant variations. Our analytical results suggest that no correlation can be determined between SI and $SiO_2$. However, an absolute correlation can be made between SI and MgO of 0.99 correlation coefficient. It is noted that MgO presence is related to the oxygen supply, which strengthens flora existence [40].

Despite their absolute element concentrations, the bottom sediments of the ponds show similar upper-crust normalized element spidergrams patterns similar to Taylor and McLennan [41] upper-crust composition (Figure 11). An exception to this is the phosphorus content, which displays a slight difference between the Greveno–Alikaina and Louka–Livadies ponds and may be related to local biological activity. As barium substitutes both K in alkali feldspars and Ca in plagioclase, and plagioclase is absent in those sediments (Table 3), Ba exhibits almost identical normalized values as K, which results in a slight negative anomaly to Ba. Moreover, Sr negative anomaly is attributed to the lack of Ca bearing minerals, such as calcite and plagioclase (Table 3). The slightly positive anomaly to Ti may be related to clay minerals and amorphous Fe-Ti oxides, which originate from the flysch of the wider area's bedrock [33]. TDS, chlorine, boron and bromine values are extremely low, due to the origin of the water, mostly deriving from snow and rainfall.

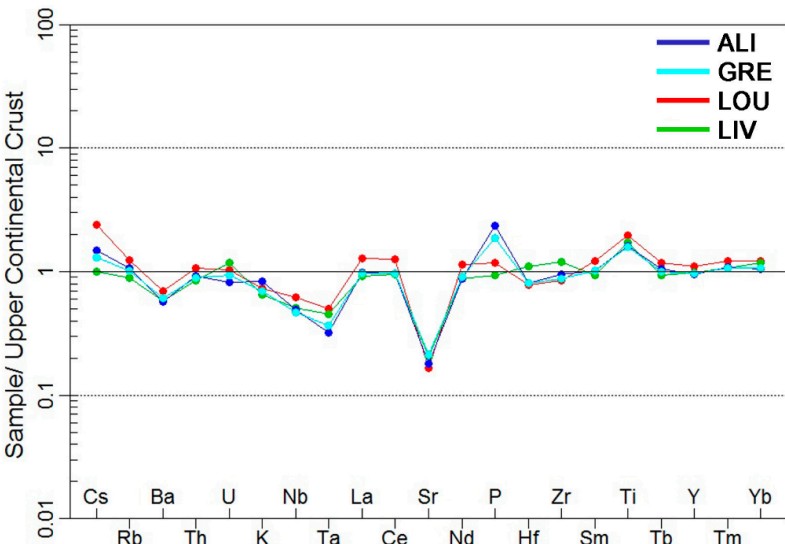

**Figure 11.** Normalized element spidergrams of MTPs from Mt. Oiti bottom sediments. Upper-crust normalization values for bottom sediments are from Taylor and McLennan [41].

The higher $PO_4^{3-}$ content recorded in Alikaina water samples is well-matched with the higher $P_2O_5$ content in the bottom sediments of the pond (Tables 4 and 5). The extremely low calcium and magnesium content of the water of all ponds can be explained by the absence of any carbonate minerals in the bottom sediments.

Kissoon et al. [42] have worked on shallow lakes in Minnesota, and they have suggested the extent to which multiple elements in shallow lake waters and sediments were influenced by a combination of

variables, including sediment characteristics, lake morphology and percent of land cover in watersheds. These data may be informative for illustrating the extent of functional connectivity between shallow lakes and adjacent lands within these lake watersheds. In contrast, the chemical results of the studied water samples from the MTPs of Mt. Oiti suggest that it is of meteoric origin, directly derived from precipitation (rain and snow), with very low mineralization (TDS < 1000 mg/L). During the two-years and five seasonal periods (wet/dry) of water sampling, the slight variations of pH values were not correlated with seasonality. Low concentrations of TOC, $NO_3^-$ and $NO_2^-$ indicate that fauna does not impact the quality significantly. However, increased amounts of nitrates during the dry period of 2013 may be attributed to the sporadic presence of animals. The most significant features, distinguishing Oiti's pond waters from any other province water bodies, are the extremely low content of any studied cation. According to Vasilatos et al. [43,44] and Megremi et al. [45,46], TDS and major and trace cations concentrations of groundwater and river water samples from the neighboring area of eastern Sterea Hellas (central Greece) are significantly higher (by several orders) than the ponds samples from Oiti. It is notable that in the wider area of the Louka pond, there is an abandoned bauxite mine. However, the bauxite buried deposits do not affect the hydrochemistry of the pond. The low-temperature conditions of the studied areas may be unfavorable for active water–rock interaction [45]. Furthermore, recent efforts for the establishment of the plant species at the pond of Louka have proved that the ecological niche of the endemic plant may include the bioclimate at altitudes as low as 1150 m. The fact that there is a trend of surface area loss for lower elevation ponds (under 2500 m) [38] is promising for the endemic plant *Veronica oetaea*.

The normalized element spidergrams of MTPs water samples from Mt. Oiti (Figure 12) indicate that all elements exhibit lower values than the wetland water with the exception of K and Rb. The observed positive anomaly of those elements could be related to the mineralogical composition of the bottom sediments (Table 3). K and Rb are very mobile elements in environmental conditions and exhibit similar geochemical behavior. It is noted that K may constitute the main nutrient ion for the performance of flora. The Sr, Fe and Si negative anomalies observed, may be attributed to water-rock interaction controlling the chemistry of the wetland water [45] that has been used for normalization. That fact may be related to the enrichment of those elements in the wetland water; normalizing (dividing) with their values results the low quotients plotted in the diagram.

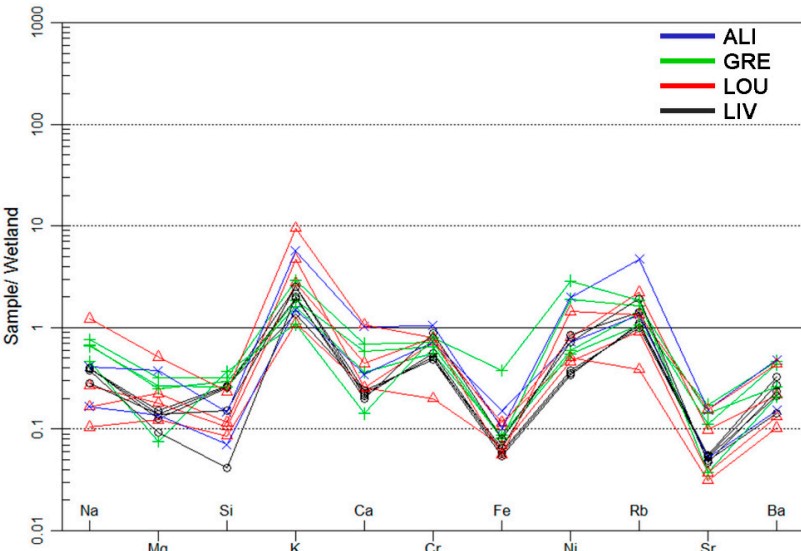

**Figure 12.** Normalized element spidergrams of MTPs from Mt. Oiti water bodies. Wetland normalization values for waters are from groundwaters of temperate humid climate zones in Europe, North America and Asia, plotted in ascending order according to their atomic number (normalization values from Shvartsev [45], presented in Supplementary Table S4).

Piper plots of the ponds water major ions contents revealed that MTPs of Mt. Oiti have similar characteristics to typical shallow and fresh water [47–49], mostly of calcium bicarbonate dominant (Figure 13). The water samples of the ponds, as can been seen in Piper diagram plot, are of calcium-bicarbonate type except the pond of Louka water sample during the dry period which is of mixed-bicarbonate type (Figure 12). According to Berner and Berner [47], calcium bicarbonate dominant water type is a common type of meteoric common water. In the dry period the Mg$^{2+}$ trend is higher compared to the wet season, probably due to dilution in the wet season and evaporation during the dry season. Moreover, it is noteworthy that according to Shvartsev [48] the wetlands around temperate humid climate zones in Europe, North America and Asia are of calcium-bicarbonate (to mixed) type, same as water samples from the MTPs of Mt. Oiti.

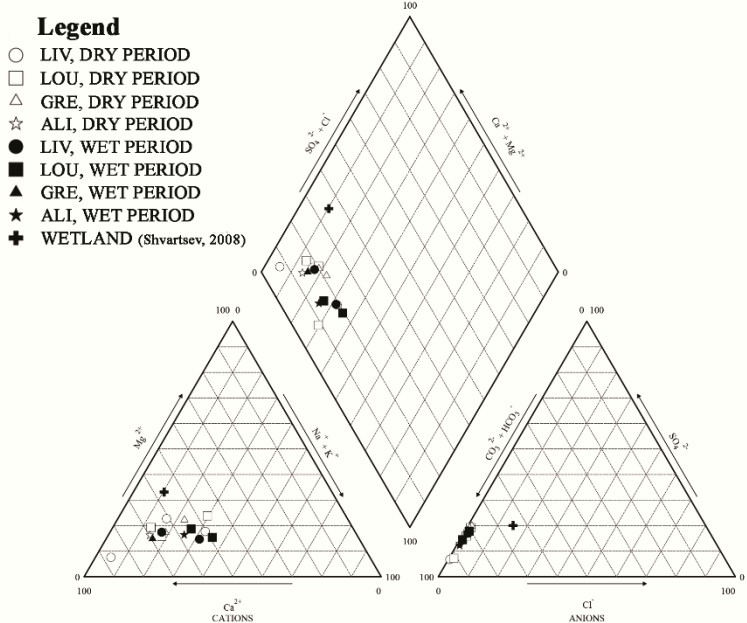

**Figure 13.** Piper diagram [49] of water samples from the MTPs of Mt. Oiti, Greece. The relevant chemical data from groundwaters of wetlands from temperate humid climate zones in Europe, North America and Asia from Shvartsev [48] is plotted for comparison.

The geochemical and mineralogical study of bottom and water pond samples indicates that the MTPs of Mt. Oiti are not exposed to any significant pressures. However, during the dry period of 2013, the presence of pasture animals was identified, as increased nitrate values were determined at the water pond samples of Louka and Greveno. Moreover, slightly increased lead values in bottom sediments at the Louka pond are probably attributed to the hunting activities (e.g., Reference [50]), as buckshot disposal was identified in situ. Furthermore, in all Oiti ponds in situ observations were conducted for evidence of tourism and other activities (Table 6). Concerning the Louka pond, the evident trampling is mainly due to the fact that it is located among summer shepherd establishments, and secondarily due to the better accessibility of the pond area because of its lower altitude. Touristic activities, such as off-road vehicles crossing, have been noticed at the peripheral belt of the ponds. Trampling by off-road vehicles may change the geomorphological character of the ponds, sediment substrate and column and subsequently prevent the germination and the development of the endemic pond species [11].

**Table 6.** Table featuring threats of study MTP area of Mt. Oiti (GR2440007) in comparison to an extensive existing list on Greek MTPs (adapted from Dimitriou et al. [27]).

| NATURACODE | WASTE DISPOSAL * | TOURISM | AGRICULTURE | HYDROLOGIC DISTURBANCE | OVERGRAZING | EXCAVATION | NO THREAT | ECOLOGICAL STATUS |
|---|---|---|---|---|---|---|---|---|
| GR1430004 | | | | | | | √ | excellent |
| GR2440002 | √ | √ | √ | | | | | at-risk |
| GR4110001 | | √ | | | | | | good |
| GR4110002 | | | | | | √ | | at-risk |
| GR4110003 | | | | | | √ | | at-risk |
| GR4110004 | | | | √ | | √ | | at-risk |
| GR4210004 | | | | √ | | | | medium |
| GR4210007 | | √ | | | | | | medium |
| GR4210008 | | | | √ | √ | | | medium |
| GR4220005 | | | √ | | | | | at-risk |
| GR4220006 | | √ | √ | | | | | good |
| GR4220007 | | | √ | | √ | | | good |
| GR4220010 | | | | √ | | | | at-risk |
| GR4220014 | √ | | √ | √ | | | | at-risk |
| GR4340001 | | | √ | | | √ | | medium |
| GR4340002 | | √ | √ | | | √ | | medium |
| GR4340010 | | √ | | | √ | | | medium |
| GR4340013 | | | √ | | √ | | | excellent |
| GR2440007 | √ | | | | √ | | | good |
| *Livadies* | | √ | | | | | | *excellent* |
| *Louka* | √ | √ | | | √ | | | *medium* |
| *Greveno* | | √ | | | | | | *excellent* |
| *Alikaina* | | √ | | | | | | *excellent* |

* Waste Disposal includes hunting wastes.

All the acquired geological, geochemical, hydrochemical and mineralogical data were combined in order to identify the pressures that affect the ecological status of MTPs of Mt. Oiti. According to the evaluation method of the ecological status by Dimitriou et al. [27], the MTPs site of Mt. Oiti was classified in accordance with the four quality categories (excellent, good, medium and at-risk), given the situation of each pond (Table 6). Herein, the ecological status represents the pressures imposed on the habitat, excluding the evaluation of flora composition data at each site [22,51]. The resulting ecological status is combined with the ecological status of the MTP sites in Greece by Dimitriou et al. [27], as shown in Table 6. Of the examined MTPs, Livadies, Alikaina and Greveno pond present an excellent eco-status meaning that these ponds tend to maintain their typical characteristics. The Louka pond presents a medium ecological status, indicating the degradation of pond area related to human and animal pressure activities.

The Louka pond is exposed to limited threats which may affect soil and water quality and the biota. Notably, these results are in agreement with the floristic study of the ponds by Delipetrou et al. [22,51]. The conservation status of the MTPs based on the floristic composition and structure (abundance and spatial distribution of the typical habitat species and intrusion of non-typical species) is excellent in Greveno, Livadies and Alikaina, but degraded in Louka where terrestrialization of the habitat is evident. Overall, the sound conservation status of the MTPs is related to the abundance and diversity of the typical habitat species which, being restricted to this habitat, increase gamma (landscape) diversity.

## 5. Conclusions

This study presents for the first time a full dataset of the geo-environmental parameters of the EU Priority Habitat Type "Mediterranean Temporary Ponds" (MTPs), along with their current ecological status in Mt. Oiti, Greece. These temporary ponds display unique ecological characteristics. The results of this study suggest that they are at a sound conservation status and have good future prospects, although, one pond is exposed to limited threats that could affect sediment and water quality and also the biota. The main pressures identified for those habitats were grazing and touristic activities.

Mediterranean Temporary Ponds of Mt. Oiti correspond to shallow depths (5–20 cm), small size (112–607 m$^2$) and high altitude (between 1000–2000 m a.s.l.). Electrical Resistivity Tomography measurements identified synforms outlined under the ponds. However, topography does not always adopt these synforms, mostly due to erosion procedures. The most significant features, distinguishing these pond waters from any other province water bodies are the extremely low content of all studied cations. The water bodies of the studied MTPs are of biocarbonate dominant type, and a fresh meteoric water origin is suggested. The bottom sediments of the ponds show similar element spidergrams patterns close to upper-crust composition. Neither bottom sediments nor water chemistry presented evidence for interaction between bedrock, sediments, and water—probably due to the low-temperature conditions of the area. Shape complexity (SI) of the MTPs of Mt. Oiti, which is an index for species richness, indicates the direct correlation of SI with organic matter, clay minerals and magnesium presence.

Since lakes and ponds are considered as early indicators of climate change, in which high altitude ecosystems are especially vulnerable, this study will allow the establishment of a geo-environmental baseline level of these habitats that can be used in future comparisons. The overall results contribute to a better understanding of the presence of temporary ponds and their development in Mediterranean environments.

**Supplementary Materials:** The following are available online at http://www.mdpi.com/2073-4441/11/8/1627/s1, Table S1: Quality control of soil analysis, Table S2: Quality control of water analysis, Table S3: Correlation coefficient matrix for soil data. Soil data per factor include data from all four MTPs at Mt. Oiti (i.e., Livadies, Louka, Greveno and Alikaina), Table S4: Wetland water normalization values used in the manuscript (data from Shvartsev, [45]).

**Author Contributions:** C.V. is accountable for the integrity of the data, analysis, and presentation of findings as a whole; M.S., K.G., J.A., E.V., P.D. and C.V. have designed the research; C.V. and M.A. did the writing—original draft preparation; C.V., J.A., E.V., S.D., P.D., K.G. and M.S. have performed fieldwork and sampling; M.A., S.P., S.A. and C.V. have performed the mineralogical, geochemical and hydrochemical analytical laboratory work; J.A., S.D. and E.V. have performed the near-surface geophysical survey the ERT data interpretation, and performed the detailed geological mapping and remote sensing data processing (GNSS and satellite imagery); P.D. and K.G. have performed the ecological survey.

**Funding:** This work was supported by the EU frame of LIFE11 NAT/GR/1014 "FOROPENFORESTS". The APC was funded by the Special Account for Research Grants, National and Kapodistrian University of Athens.

**Acknowledgments:** The authors would like to thank Spyridon Mavroulis, Dimitris Michelioudakis, George-Pavlos Farangitakis, Georgia Mitsika, Eleni Kaplanidi, Katerina Polykreti and Alexandra Zavitsanou for their help during the acquisition of the geophysical measurements. The two anonymous reviewers are kindly thanked for their constructive suggestions and comments to an earlier draft of this manuscript.

**Conflicts of Interest:** The authors declare no conflict of interest. The funders had no role in the design of this work; in the collection, analyses, or interpretation of data; in the writing of the manuscript, or in the decision to publish the results.

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
