# Peer review of "Assessment of the Geo-Environmental Status of European Union Priority Habitat Type “Mediterranean Temporary Ponds” in Mt. Oiti, Greece"

_water, doi:10.3390/w11081627_

Round 1
Reviewer 1 Report
The study is aimed at the assessment of the geo-environmental conditions of Mediterranean Temporary Ponds in Mt. Oiti, Greece. While the presented study delivers engaging results, some issues need to be addressed before its prospective publication.
The introduction is very vague, describes various aspects of the protection of vernal ponds, but does not introduce the reader to the geochemical issues raised in the article. Authors claim that their paper is the "first integrated study which contributes to the understanding of the ecosystems of Greek MTP areas, by investigating a wide range of parameters, which determine the ponds' geological background and ecological status". I doubt whether the study of four objects located at a distance of at most several kilometres and located in similar geological conditions (as suggested in Fig. 2) can contribute to the implementation of an ambitious work objective.
Moreover, the research did not include biota or dynamics of water and trophic conditions. In the understanding of the problem, it is not helpful to determine the content of several dozen of chemical elements, in large part of an undefined ecosystem role. Especially that these data have not been analyzed comprehensively and have not been used in the Discussion
Lines 46-47 there are certainly more rare or "uncommon" species than just 17
Line 54 I doubt whether an object of 1m2 could be considered as a "pond."
Please complete description of bottom sediments of Graveno MTP (Table 2)
Line 428 Authors wrote that MTPs of Mt Oiti have typical "shallow and fresh ground waters". This statement is in opposition to previous remarks about the supply of MTPs by rainwater and snowmelt (Line 401)
The discussion needs editing to make the manuscript's intent more clear. In the Discussion show the principles and/or relationships found in the Results. Show where there might be exceptions to these principles, and compare the results and your interpretations of results with findings in the literature. Sometimes it is helpful to follow the order of topics in the Results in the Discussion, but there is a risk in repeating the Results in the Discussion. How important are your findings for understanding (protection) of MTPs?
Given the above, my evaluation is that the manuscript should be considered for publication after major revision.
Reviewer 2 Report
General remarks:
Please improve grammar and in particular punctation and syntax in the manuscript.
Reorganize paragraphs and assign them clearly to the appropriate section.
Why is the study important for the journal’s readership?
Broaden the perspective from a case study to general findings.
Abstract:
The abstract is puzzling at the current state. First you introduce generally, then the study area and the methods are described, and then you jump back to the study aims.
I would suggest to reorganize the abstract in a logical manner: general intro, study aims, study area and methods, results and conclusions.
Keywords: It would help the journals readership to provide the full name “Mediterranean Temporary Ponds” as keyword here instead of “Mountain ponds” and “MTPs”; when “ecology” is being written in lower case letters the same must be applied for “Environmental Geochemistry” …
Line 47: „uncommon endemic“? I would suggest to delete uncommon.
Line 48-49: I don’t understand the term “rare” in this context and how a habitat with an ephemeral character can be a most long-term of aquatic habitats. Please clarify.
Line 49: Don’t start a sentence with an abbreviation. Correct to “European” and correct elsewhere.
Line 51-55: What indicators for what kind of monitoring. Be more precise.
Line 62: Place a dot after “specialized” and start a new sentence.
Line 64: change to “… a considerable flora and fauna species diversity.”
Line 67: Delete “Globally” at this point.
Line 78: I think it may help the journals readership to explain the term “volcanoclastic”.
Line 86-88: Isn’t ecosystem function meant at this point?
Line 97: What is the reason for the site selection - these particular MTPs?
Line 100: ponds
Line 100: Please write the species name in scientific style (italics and Authors name).
Line 125: What is meant with “traditional”?
Line 126: Aren’t these species characteristic? Species names must always be written in Italics.
Section 2.1.: Please inform the reader about the reason for the site selection.
Line 135: When mentioning geological fieldwork including mapping and sampling first, I would expect some explanation about these methods first. I would suggest to reorganize the paragraphs of the section.
Line 207-210: Reorganize the sentence and delete the redundant opening words.
Line 210-212: The sentence is incomprehensible. Reorganize the syntax and add missing words.
Line 214-215: Again, reorganize the sentence, eventually spilt in two.
Line 235: I would suggest to delete “Moreover” since it is redundant here.
Figure 4: Figure 4 can be regarded as redundant. It does not provide any explanatory information.
Figure 5, 6, 7, 8: The image resolution is poor. Please improve.
Line 277-278: Is square metre meant here? Insert blank spaces where appropriate here and elsewhere (e.g. line 287).
Line 289-290: Which method was being used to identify these human impacts for those MTPs and almost all within the National Forest Park area? Please explain: What kind of category is a National Forest Park? Isn’t it a National Parc according to the IUCN criteria? Please explain.
Line 314-317: Is this your result or do you discuss your results here? Please be clear with the appropriate sections content.
Line 329: Sentences always have to start with capital letters. Eventually change to “The pH-value varies …” delete the redundant blank space.
Line 329-334: The results are already discussed in this paragraph and elsewhere in the result section. Please clarify what are results and what is discussion and then reorganize the sentence to the appropriate sections.
Line 343: Correct the species name and use scientific style for species names.
Figure 11 caption: Delete the second “species”.
Line 401: Please insert blank spaces where needed.
Line 415: Please change “This is hopeful for…” to a more appropriate expression of what you mean.
Line 426-427: Please correct punctation.
Line 433 and 435: I would suggest to change the term “it is noteworthy” to a more appropriate term.
Figure 14 caption: Be more precise with “the chemical data…”. Delete “Also”.
Line 448: I´m not sure whether “Trampling by off-road vehicles” is the correct term to express what is meant.
Line 464-465: Why are these results in agreement with the floristic study of the ponds by Delipetrou et al. ? A discussion on this topic at this point is needed since you also come back to this statement in your conclusions.
Line 477-482: This is the first time that gamma diversity is being mentioned. I would suggest to include some words in the introduction and the discussion section when you come to this conclusion.
In addition, I would expect more information about the floristic studies and relations to the results in the discussion section.
Line 485: Insert punctation after “However”.
Round 2
Reviewer 1 Report
The authors did a significant job in improving the article, which now could be accepted for publication. However, minor editorial errors remain and some examples of awkwardness or errors I showed below. I recommend a careful reading of the manuscript and its meticulous editing.
The sentence "The main human impact ... was grazing" ... (Line 43) is somewhat awkward.
Line 54 - the word "pond" should be written with a lowercase letter.
Line 151-152 - add "above sea level (m a.s.l.) to the mountain height
e.g. Line 332 - should be "range from 112 to 608 m2"
Fig. 9 in the figure caption, please avoid the unspecified abbreviations of the names (in this case SI)
Fig. 14 the diagram does not show ion concentration, as erroneously given in the figure caption
Reviewer 2 Report
General remark:
The revised manuscript is improved and most of the suggested comments were taken in account.
I still feel that the grammar has to be improved. May be English editing by a native speaker would help. Also, the punctation is still insufficient to some extent.
In particular, the authors switch from present tense to past tense and back. I strongly suggest to follow the journals roles regarding grammar and spelling.
Moreover, I don’t understand why the authors - again - placed Figure 4 in the manuscript. I´m convinced that the journals readership is able to understand the sampling procedure from the text. There are already 8 pictures of the four sites provided in Figure 3 and I thus regard Figure 4 as redundant.
Details:
Line 42: Of “any” studied element? May be better provide information what elements were studied.
Line 43-44: “The main human impact identified were grazing and trampling by vehicles…”? Please correct the sentence and delete “t”.
Line 62: Place the “and” behind “bodies” and delete “largely”.
Line 100: Aren´t ecosystem functions meant at this point?
Line 104: I would suggest to change “wood harvest” to “forestry”.
Line 126: Please write species names always in italics.
Figure 3 caption: I would suggest to provide a-d with the names of the four sites and add “dry period left hand side; wet period right hand side” to shorten the caption.
Line 209: Delete “of”.
Line 242: Add “were” and “was” in Line 245.
Line 295: Add a behind “identified”
Line 384: “streams”
Line 385: What values? Your recorded values?
Discussion Section:
I would suggest to use the first person in particular in the discussion section in order clarify what are your results and what are these from literature.
Line 543: Place a comma behind the reference.
Line 562 and elsewhere: Use the abbreviation EU.
Line 567: Place a dot behind prospect. Start a new sentence for threats and pressures.
